

# Insects' perception and behavioral responses to plant semiochemicals

Diriba Fufa Serdo

Department of Biology, Ambo University, Ambo, Ethiopia

## ABSTRACT

Insect-plant interactions are shaped by the exchange of chemical cues called semiochemicals, which play a vital role in communication between organisms. Plants release a variety of volatile organic compounds in response to environmental cues, such as herbivore attacks. These compounds play a crucial role in mediating the interactions between plants and insects. This review provides an in-depth analysis of plant semiochemicals, encompassing their classification, current understanding of extraction, identification, and characterization using various analytical techniques, including gas chromatography-mass spectrometry (GC-MS), liquid chromatography-mass spectrometry (LC-MS), nuclear magnetic resonance (NMR) spectroscopy, and infrared (IR) spectroscopy. The article also delves into the manner in which insects perceive and respond to plant semiochemicals, as well as the impact of environmental factors on plant odor emission and insect orientation. Furthermore, it explores the underlying mechanisms by which insects perceive and interpret these chemical cues, and how this impacts their behavioral responses, including feeding habits, oviposition patterns, and mating behaviors. Additionally, the potential applications of plant semiochemicals in integrated pest management strategies are explored. This review provides insight into the intricate relationships between plants and insects mediated by semiochemicals, highlighting the significance of continued research in this field to better understand and leverage these interactions for effective pest control.

## INTRODUCTION

Plants and insects have evolved to interact with one another in complex and fascinating ways. Insect-plant interactions are shaped by the exchange of chemical cues called semiochemicals, which play a vital role in communication between organisms. A key aspect of this interaction is the use of chemical cues, both by plants to attract beneficial insects and deter pests, and by insects to locate and evaluate potential food sources (*Prasad, 2022*). The term semiochemical is derived from the Greek word "semeion", meaning a signal, and these substances vary in their molecular weights based on their carbon chain (*Soroker, Harari & Faleiro, 2015*). Semiochemicals are chemicals or mixtures released by an organism that carry messages and influence the behaviors of other organisms (*Mweresa et al., 2020*). These chemical signals play a crucial role in various biological processes, such as plant-insect interactions, insect-microbe interactions, and insect-insect interactions.

Corresponding author
Diriba Fufa Serdo,
diribafufa2@gmail.com

Insects live in an environment surrounded by numerous semiochemicals, which can originate from other insects or the host plant. Insects interact with semiochemicals from host plants, and these volatile compounds can modify the behavior of those insects.

Various plant organs contain volatile and non-volatile semiochemicals that play a crucial role in insect-plant interactions (*Pichersky, Noel & Dudareva, 2006*). Insects are attracted to a range of volatile compounds produced by plants through different metabolic pathways, including those derived from aromatic amino acids, fatty acid degradation, terpenoid biosynthesis, and specific glycosides like glucosinolates (*Städler & Reifenrath, 2009*). Additionally, plants release highly volatile compounds such as isoprene, ethylene, formaldehyde, and various organic acids (*Brilli et al., 2011*). However, the effects of these compounds on insect-plant interactions are not well understood, and only a few studies have examined their impact (*Goyret, Markwell & Raguso, 2008*). Plant semiochemicals play a crucial role in insect behavior, with pollinators using them to locate flowers for pollen and nectar uptake. Moreover, herbivorous insects use plant semiochemicals to locate suitable host plants and assess their quality, while predatory insects use them to find herbivorous prey or hosts (*Raguso, 2008*). Additionally, plant volatiles can interact with aggregation pheromones and aid insects in finding mates on host plants (*Reinecke, Hilker & Monika, 2002*). For instance, some insects use host plant chemicals as sex pheromones for mating (*Reddy & Guerrero, 2004*). Certain insects, such as butterflies and moths, rely on pyrrolizidine alkaloids derived from their host plants as a source of food to protect themselves from natural enemies (*Nishida, 2002*).

The African palm weevil *Rhynchophorus phoenicis* Fabricius (1801) utilizes a mixture of volatile esters from the host plant oil palm *Elaeis guineensis* Jacquin (1763). The presence of ethyl acetate in this mixture causes male weevils to release the aggregation pheromone rhyncophorol, attracting both sexes for mating (*Reinecke & Hilker, 2014*). Male orchid bees utilize a mixture of terpenoids from orchids as a pheromone to establish specific locations where males compete for females to mate (*Ezzat et al., 2019*). Background odors are also important in providing information on habitat quality and facilitating the effectiveness of plant volatile signals in insects (*Beyaert & Hilker, 2014*). The complex interactions between plant volatiles, pheromones, and environmental cues demonstrate the multifaceted role these chemicals play in insect chemical ecology. Researchers are exploring the diverse types of plant semiochemicals, ranging from volatile organic compounds to root exudates, and their influences on insect behaviors. By delving into the mechanisms by which insects perceive and respond to these chemical signals, this review sheds light on the intricacies of insect-plant communication that shape ecological dynamics. For scientists studying chemical ecology, entomology, or plant-insect interactions, this review provides a comprehensive overview of the current understanding of insects' perception and behavioral responses to plant semiochemicals. By highlighting the significance of semiochemical-mediated interactions in shaping insect behaviors such as foraging, mating, and oviposition, this manuscript offers valuable insights for researchers aiming to unravel the complexities of insect-plant relationships. Furthermore, understanding how insects perceive and respond to plant semiochemicals can inform the development of sustainable pest management strategies, crop protection, and conservation, and this is
relevant to a broad audience interested in sustainable agricultural practices and biodiversity conservation.

## SURVEY METHODOLOGY

### Review questions and literature search

How do insects perceive plant semiochemicals? What are some examples of the behavioral responses of insects to plant semiochemicals? What are the potential applications of understanding insect perception and responses to plant semiochemicals in pest management strategies? To ensure comprehensive coverage of the literature, a systematic search of scientific databases, including PubMed, Scopus, and Web of Science was conducted using relevant keywords such as insect-plant interactions, insects' perception, pest management, plant odor emission, plant semiochemicals, and volatile organic compounds and chemical ecology. Peer-reviewed articles, reviews, and book chapters were included in the survey to encompass a broad range of findings in the field. Additionally, cross-referencing and citation tracking were employed to identify seminal studies and recent publications that contribute to the understanding of plant semiochemicals and insect responses. By synthesizing findings from diverse studies, a holistic view of the field was presented in this manuscript. The literature review was done following the PRISMA (Preferred Reporting Items for Systematic Review and Meta-Analyses) guidelines (*Kamioka, 2019*).

### Selection criteria

Articles obtained from the initial search were screened based on defined inclusion and exclusion criteria. Inclusion criteria for conducting a review on insects' perception and behavioral responses to plant semiochemicals include studies focusing on this topic, research examining the role of pheromones and other chemicals in insect-plant communication, investigations into chemical cues used by insects, and articles exploring the effects of plant semiochemicals on insect behavior and ecology. Exclusion criteria involve studies not directly related to the topic, non-English articles, outdated or irrelevant research, and non-peer-reviewed sources.

### Screening process, data extraction and quality assessment

The screening process involves reviewing titles, abstracts, and full texts of studies to identify relevant research that meets the inclusion criteria and excluding those that do not. Data extraction involves systematically collecting key information from selected studies using a standardized form and recording details on study objectives, methods, major findings, and limitations. The quality assessment process in a review on insect perception and behavioral responses to plant semiochemicals involves using a tool, such as SYRCLE's RoB tool, to evaluate articles based on criteria like study design and methodology (*Basha & Mamo, 2021*). Scores are assigned for each question, with a total score ranging from 0 to 7. Articles with scores of 6–7 are considered ''high quality'', 3–5 as ''medium'', and 0–2 as ''low quality''. Studies of ''medium'' and ''high'' quality are considered for analysis, with no exclusions based on low-quality scores in this case. The results of the quality assessment

help in interpreting findings, addressing limitations, and informing recommendations in the review.

## RESULTS

### Categories of plants' Semiochemicals

#### Plant hormones

Plant hormones are crucial in controlling various developmental processes and stress response pathways, enabling plants to adapt to a variety of environmental challenges, from living organisms to physical conditions (*Bari & Jones, 2009*; *Hu et al., 2024*). Plant hormones, such as jasmonic acid and salicylic acid, play crucial roles in regulating plant responses to insect herbivory. These hormones can induce the production of defense compounds and signaling molecules in response to herbivore attacks. Plant hormone signaling pathways are intricately connected and influenced by a complex network of defense and developmental processes. Understanding how plants integrate multiple hormonal signals in response to various environmental and developmental cues is a significant challenge. It is crucial to recognize that the nature and response of plant interactions with stressors vary depending on the specific plant-pathogen system, timing, hormone concentration, and tissue location (*Bari & Jones, 2009*). Future research on plant hormones and insect-plant interactions should focus on deciphering hormone roles in defense responses, elucidating molecular mechanisms, engineering defense responses, developing diagnostic tools, and investigating tritrophic interactions. This will advance our understanding of plant-insect interactions and inform the development of novel pest management strategies.

#### Allelochemicals

These are chemicals produced by plants that influence the growth, survival, or behavior of other organisms, often acting as defense mechanisms against herbivores, pathogens, or competing plants. There are also herbivore-induced plant volatiles which are released by plants in response to phytophagous damage, serving as a warning signal to nearby plants and attracting natural enemies of the phytophagous insects for biological control. These includes terpenes, green leaf volatiles, benzyl acetones and others. There are various types of plant allelochemicals, each with different effects on target organisms. These are basically categorized into (i) kairomones: that benefits the receiver but not the emitter whereas (ii) allomones: benefit emitter/producer only and (iii) synomones: that support both the producer and the beneficiary (*Kost, 2008*; *Mweresa et al., 2020*). For example, flowers produce floral scents that attract pollinators, such as bees and butterflies, benefiting both the plant (through pollination) and the pollinator (through nectar as a food source). Pest-infested plants may produce volatiles to attract natural enemies of the same pest in what is termed "a cry for help" (*Dicke et al., 2009*; *Hung et al., 2015*). The study on allelochemicals has revealed complex interactions between plants and insects. Despite progress in understanding allelochemicals and its role in insect-plant interactions, several areas remain underexplored. These include mechanisms of perception, chemical synthesis and degradation, long-term effects, multiple stressors, co-evolution, cross-species

interactions, microbial mediation, behavioral adaptations, and economic and social implications.

### Phytochemicals

Phytochemicals are a class of constitutive metabolites, which play a crucial role in the survival and proper functioning of plants. These bioactive compounds are produced by plants to overcome environmental stress and regulate essential physiological processes, including growth and reproduction (*Molyneux et al., 2007*). They are found in various plant tissues, including stems, leaves, roots, seeds, fruits, and flowers, with many being concentrated in the outer layers of plant tissues (*King & Young, 1999*; *Rabizadeh et al., 2022*). Phytochemicals can be categorized into primary and secondary metabolites based on their function in plant metabolism. Primary metabolites are essential for plant life and include carbohydrates, amino acids, proteins, lipids, purines, and pyrimidines of nucleic acids. They are the fundamental building blocks of plant growth and development, and play a pivotal role in plant defense against insect herbivores. During an insect attack, primary metabolites such as carbohydrates and amino acids are mobilized and redirected within the plant to provide substrates for local defense responses. The import of systemic resources from other parts of the plant can also occur, enabling the production of defense-related compounds. These compounds, including cinnamic acid and phenolic glycosides, contribute to the plant's defense against insect herbivores (*Zhou et al., 2015*).

Secondary metabolites are produced through metabolic pathways derived from primary metabolic pathways, and they function as defense compounds, providing protection against herbivores and pathogens. Additionally, these secondary metabolites act as signal compounds, attracting pollinators and seed dispersers to facilitate the plant's reproductive processes (*Hussein & El-Anssary, 2019*; *Divekar et al., 2022*; *Wink, 2003*). Secondary metabolites are classified into three main groups based on their biosynthetic pathway: nitrogen-containing compounds (*e.g.*, alkaloids, glucosinolates, and cyanogenic glycosides), phenolic compounds (*e.g.*, phenylpropanoids and flavonoids), and terpenes (*Jamwal, Bhattacharya & Puri, 2018*; *Jan et al., 2021*). Alkaloids are a class of nitrogen-containing compounds that are produced in response to environmental stimuli and exhibit remarkable biological activities and structural diversity (*Ng, Or & Ip, 2015*; *Takooree et al., 2019*). Alkaloids play a crucial role in plant defense, serving as a natural deterrent against herbivorous insects and pathogens. These compounds possess toxic properties that render them unpalatable or even deadly to these organisms, thereby impeding their ability to feed on or infect the plant. Additionally, alkaloids can repel pests such as aphids, whiteflies, and other insects that feed on plant tissues, providing an additional layer of protection against damage and disease.

Glucosinolates, a class of compounds mainly found in the Brassicaceae family, play a crucial role in insect-plant interaction. These compounds are synthesized in plant cells and stored in various plant tissues. They can serve as attractants for beneficial insects such as bees and butterflies (*Giamoustaris & Mithen, 1996*). Notably, over 25 insect species from the Coleoptera, Lepidoptera, and Diptera orders have been found to be attracted to and utilize glucosinolates as a primary resource (*Hopkins, Van Dam & Van Loon, 2009*).

Furthermore, researchers have discovered that glucosinolates have a profound impact on a broad range of generalist herbivores, deterring them from feeding on the plant and providing a natural defense against herbivory (*Hopkins, Van Dam & Van Loon, 2009*; *Kos et al., 2012*). Cyanogenic glycosides are amino acid-derived plant compounds that exhibit widespread distribution across over 100 families of flowering plants (*Francisco & Pinotti, 2000*; *Rabizadeh et al., 2022*). In particular, cyanogenic glycosides serve as a deterrent to herbivores by releasing HCN, which is toxic to these insects and thereby helps protect the plant from damage caused by herbivory (*Nyirenda, 2020*). This defense mechanism is crucial for the survival of plants, allowing them to adapt to their environment and compete with other organisms for resources.

Phenolic compounds are a crucial component of plant defense responses, playing a vital role in the production of phytoalexins, which are toxic compounds that help plants resist pathogens and insects. These compounds serve as antimicrobial and antioxidant agents, enabling plants to evade pathogenic infections and protect major tissues from damage. The biosynthesis of phenolic compounds occurs through various pathways, including the shikimate, pentose phosphate, and phenylpropanoid pathways (*Lin et al., 2016*; *Pratyusha, 2022*). The phenolic compound family encompasses a range of substances, including flavonoids, phytoalexins, curcumin, resveratrol, epigallocatechin-3-gallate, and others (*Kennedy & Wightman, 2011*).

Terpenoids are the most abundant group of plant secondary metabolites produced in flowers, vegetative tissues, and roots (*Dudareva, Pichersky & Gershenzon, 2004*). Terpenoids play a crucial role in plant-insect interactions, serving as chemical mediators that facilitate communication between plants and insects. These compounds can serve as defense mechanisms against herbivorous insects, acting as repellents to deter them from feeding on the plant or attracting predators that feed on the herbivores. Additionally, terpenoids can be induced to be emitted as volatile organic compounds in response to insect attack, attracting predators or parasitoids that feed on the herbivores and providing indirect defense against the herbivores. Furthermore, terpenoids can act as attractants for beneficial insects, such as pollinators and predators, which can help plants defend against herbivores. Moreover, they can facilitate communication between plants and insects, allowing plants to send signals to insects about the presence of herbivores or other threats (*Boncan et al., 2020*; *Sharma, Anand & Kapoor, 2017*). Although significant advancements have been made in elucidating the complex interactions between phytochemicals and insects, several aspects of this relationship remain underserved by existing research. Specifically, the impact of phytochemicals on insect microbiome, social behavior, developmental biology, migratory patterns, population dynamics, human-insect interactions, plant-soil interactions, and climate-induced changes has yet to be comprehensively explored.

### Root exudates

Plant roots secrete a complex mixture of molecules known as root exudates, which are composed of thousands of different substances, including organic acids, amino acids, fatty acids, sugars, mucilage, phenolics, proteins, and more (*Bais et al., 2006*; *Dennis, Miller & Hirsch, 2010*; *Steeghs et al., 2004*). Several signaling chemicals have been

identified, including ethylene, strigolactones (SLs), jasmonic acid (JA), (-)-loliolide, and allantoin (*Wang et al., 2021*). These root-secreted signaling chemicals play a crucial role in insect-plant interactions and defense against insect herbivores. Ethylene is a classical phytohormone that responds to plant neighbors, herbivores, pathogens, or other attackers (*Baldwin et al., 2006*; *Farmer, 2001*), while JA is a common signaling chemical that elicits the production of defensive metabolites in plants against feeding herbivores or plant competitors (*Kong et al., 2018*; *Martínez-Medina et al., 2017*). (-)-Loliolide, a carotenoid metabolite, has been found to be the most ubiquitous monoterpenoid lactone in plant families (*Grabarczyk et al., 2015*) and may act as a signaling chemical in plant defenses against pathogens (*Pan et al., 2009*) and herbivores (*Murata et al., 2019*).

Root exudates have been shown to play a crucial role in plant defense against insect herbivores by producing chemical signals that attract predators or parasitoids that feed on herbivores, thereby providing indirect defense (*Khashiu Rahman, Zhou & Wu, 2019*). Additionally, root exudates can stimulate the emission of volatile organic compounds, which attract beneficial insects or repel herbivores, while also influencing the composition of the rhizosphere microbiome and its impact on plant defense. Furthermore, root exudates facilitate plant-microbe interactions, enabling microorganisms to produce compounds that are toxic to herbivores or enhance plant defense mechanisms. Finally, root exudates can contain species-specific signals that convey information about local conditions and influence plant defense against insect herbivores. Despite advances in the study of root exudate, the key aspects remain poorly understood. The chemical composition, timing, and spatial distribution of exudates, as well as insect detection mechanisms have not been thoroughly investigated. Microbial influences on exudate chemistry and functional significance of interactions are also unknown. To advance knowledge, interdisciplinary research is needed to address these gaps and integrate expertise from plant physiology, ecology, entomology, microbiology, and analytical chemistry.

## Extraction, identification and characterization of semiochemicals

The primary methods for isolating, identifying, and optimizing plant semiochemicals involve conventional and reverse chemical ecology approaches (*Barbosa-Cornelio et al., 2019*; *Stökl & Steiger, 2017*). Extracting, identifying, and characterizing semiochemicals, involves several steps. The first step is to collect samples containing the semiochemicals of interest. This could involve collecting plant tissues, insect secretions, or environmental samples from the field or conducting controlled experiments. Once the samples are collected, the next step is to extract the semiochemicals from the samples. Extraction methods can vary depending on the nature of the semiochemicals and the sample matrix. Common extraction techniques include solvent extraction, solid-phase microextraction (SPME), or headspace extraction for volatile compounds (*Barbosa-Cornelio et al., 2019*; *Maksimovic et al., 2017*).

After extraction, the extracted compounds need to be analyzed to identify and quantify the semiochemicals present. Analytical techniques such as gas chromatography-mass spectrometry (GC-MS), liquid chromatography-mass spectrometry (LC-MS), nuclear magnetic resonance (NMR) spectroscopy, or infrared (IR) spectroscopy can be used for

compound identification (*Fraser, Mechaber & Hildebrand, 2003*; *Mweresa et al., 2020*; *Qiu et al., 2004*). Moreover, Gas chromatography-electroantennographic detection (GC-EAD) is being widely used to screen chemical mixtures of plant volatile compounds and insect pheromones (*Johnson et al., 2020*; *Munro et al., 2020*). GC-MS is a technique used for compound identification and quantification, while GC-EAD is used to detect bioactive compounds that induce an electrophysiological response in insects, which can help in the screening of plant volatile compounds and insect pheromones. The GC-MS technique is particularly suitable for analyzing low-molecular mass and mid- to low-polarity compounds (*Pocsfalvi et al., 2016*).

Once the compounds are separated and detected by the analytical instrument, their identities need to be established. This can be done by comparing the mass spectra, retention times, or other characteristic properties of the compounds with those of known standards or databases. Characterization of semiochemicals involves determining their biological activity, role in chemical ecology, and interactions with other organisms. This can be done through bioassays or behavioral experiments to assess the behavioral responses of target organisms to the identified compounds. If the exact structure of the semiochemical is unknown, structural elucidation techniques such as fragmentation analysis in mass spectrometry, nuclear magnetic resonance (NMR) spectroscopy, or other spectroscopic methods can be used to determine the chemical structure.

Given that natural semiochemicals are typically emitted in limited amounts, chemical synthesis is utilized to produce sufficient quantities of these compounds. This involves generating and utilizing synthetic analogs (*Maksimovic et al., 2017*). In recent times, the reverse chemical ecology approach has gained favor due to advancements in understanding the molecular underpinnings of insect olfaction. This methodology revolves around screening specific chemosensory proteins responsible for detecting semiochemicals and is considered a contemporary technique for pinpointing active volatile semiochemicals (*Barbosa-Cornelio et al., 2019*). The extraction, identification, and characterization of plant volatiles or semiochemicals are essential for comprehending ecological interactions and communication among organisms. This knowledge assists in the development of pest management strategies, manipulation of insect behavior, and discovery of novel compounds for agricultural and conservation purposes.

Understanding how insects respond behaviorally to active semiochemicals is pivotal in their recognition (*Ezzat et al., 2019*; *Faleiro et al., 2014*). Semiochemicals have been utilized in pest control for over a century, effectively managing pests like *Ectomyelois ceratoniae* Zeller (1839), *Tuta absoluta* Meyrick (1917), and *Spodoptera frugiperda* J.E. Smith (1797) (*Soroker, Harari & Faleiro, 2015*). However, the success of using these in pest control programs can be influenced by various factors, including biological divergences in species' mate-finding behaviors, the varied chemical makeup of pheromones, economic and political regulations pertaining to pheromone usage in different regions, and the deployment of controlled-release dispensers, appropriate trap designs, and densities (*Nishida, 2002*).

## Insects' behavior and perception in response to plants semiochemicals

Insects perceive plant semiochemicals through their chemosensory systems, which include olfactory receptors and gustatory receptors. These receptors enable insects to detect and respond to various chemical compounds released by plants, such as volatile organic compounds. By interacting with these semiochemicals, insects can make decisions about their behavior, such as finding food sources, locating mates, or avoiding predators (*Yang et al., 2019*; *Zwiebel & Takken, 2004*). Insects have developed a highly sensitive sense of smell through the presence of odorant receptors (ORs), ionotropic receptors (IRs), and sometimes gustatory receptors (GRs) in their olfactory receptor neurons (ORNs) (*Andersson, Löfstedt & Newcomb, 2015*).

Olfactory receptor neurons are primarily located in sensilla on structures like antennae and maxillary palps (*Benton, 2009*), where they respond to chemical stimuli and transmit signals to the insect's brain for further processing (*Leal, 2013*). The detection and integration of semiochemical signals also involve the expression of chemosensory proteins like odorant-binding proteins (OBPs) and pheromone-binding proteins (PBPs) in the antennae, which are crucial for odorant transport and recognition (*Leal, 2013*). The success of insect orientation towards plant volatiles is influenced by a combination of sensory capacities and phenotypes, including factors like physiological state, experience, and motivation (*Anton, Dufour & Gadenne, 2007*). Additionally, abiotic factors such as temperature, wind conditions, humidity, light intensity, and circadian rhythms play a role in shaping host plant odors and influencing insect responses to semiochemical signals (*Gouinguené & Turlings, 2002*). These factors highlight the complex interplay between environmental conditions, plant volatiles, and insect behavior in the perception of semiochemicals (*Saunders, 2012*).

### Environmental factors influencing plant odor emission and insect orientation

Plant odors, referred to as 'volatile packages', contain behaviorally important compounds and can be influenced by several biotic and abiotic factors. Biotic factors include presence of herbivores, microbial interactions and competition with neighboring plants, while abiotic factors are temperature, light, humidity, wind speed and direction (*Reinecke & Hilker, 2014*). The diversity of plant species in a particular habitat can impact how insects locate resources by affecting the filtering of volatile compounds that indicate the presence of resources from a complex background of odors (*Randlkofer et al., 2010*). For example, insects like bark beetles may avoid volatiles emitted by plants that are not suitable hosts when searching for appropriate hosts (*Zhang & Schlyter, 2004*). However, the overall odor of the habitat can also play a role in guiding insects towards key volatiles, with plant odors potentially masking the scent of host plants and influencing orientation. The structure of vegetation, including the number and size of plants, can impact how odors spread and how insects navigate towards them (*Randlkofer et al., 2010*).

Plant odors can resemble pheromone plumes in terms of containing behaviorally important compounds in varying sizes and quantities (*Beyaert & Hilker, 2014*). The quality and quantity of plant odors can be influenced by biotic factors such as pathogens and

herbivores. Changes in plant odors induced by stress, such as herbivory or egg deposition, can be observed in both leaf and root emissions (*Hilker & Meiners, 2010*). Additionally, the scent of floral parts of a plant can change in response to factors like pollination. The ways in which insects respond to plant odors can vary based on the specific characteristics of the insect perceiving the scent. This adaptability in insects' responses to plant odors highlights their ability to adjust to changes in their environment. While significant research has been conducted on the relationship between environmental factors and plant odor emission and insect orientation, several gaps remain unclear. These include the effects of high-altitude and high-latitude environments, desert and arid ecosystems, urban pollution, and the interactions between multiple environmental factors. Long-term studies and species-specific responses also needs further investigation to comprehensively understand these complex relationships.

### The influences of experience on insects' behavior and responses to plant odors

The sensory and olfactory recognition of host plants in insects is influenced by their past experiences, enabling them to link scents with either beneficial resources or negative consequences (*Gupta & Stopfer, 2011*). While previously believed to be primarily essential for insects with diverse diets, the ability to associate plant odors is also critical for insects that specialize in feeding on specific plant species. The impact of odor encounters on insect behavior can manifest rapidly or persist over extended periods (*Gupta & Stopfer, 2011*). According to the Hopkins host selection principle, insects tend to prefer the odors of host plants they encountered during their larval stage once they reach adulthood (*Schoonhoven, Van Loon & Dicke, 2005*), although the transferability of this preference across developmental stages is a subject of debate. Conversely, the Neo-Hopkins host selection principle suggests that odors experienced post-pupal stage can influence adult preferences. Recent research indicates that experiences during the larval stage can affect the adult insects' responses to odors (*Blackiston, Casey & Weiss, 2008*). The impact of experience on insects' behavioral responses to plant odors has been extensively investigated. However, several aspects like individual variability in plasticity, long-term effects, interactions between sensory modalities, and the evolutionary pressures shaping experience-dependent plasticity require further exploration to fully understand its mechanisms and implications.

### Sick insects and their responses to plant odor

Infection of insects by entomopathogens or infestation by parasites can induce changes in their phenotype, potentially impacting their olfactory responses. For instance, *Mallon, Brockmann & Schmid-hempel (2003)* found that honey bees exhibited reduced odor learning abilities following immune challenge, and *Schütte et al. (2008)* observed decreased attraction of the predatory mite *Phytoseiulus persimilis* Athias-Henriot (1957) to plant volatiles when infected. Another study showed that, infection with bacteria and fungi can lead to changes in gene expression of odor-binding proteins in *Anopheles gambiae* mosquitoes (*Aguilar et al., 2005*). Infection with the bacterium *Wolbachia sp.* has been shown to influence insect behavior and fitness (*Werren, Baldo & Clark, 2008*), while

*Peng & Wang (2009)*, *Vosshall & Hansson (2011)* demonstrated that *Wolbachia sp*. infection can change olfactory behavior and gene expression in Drosophila flies. Parasitization of *Manduca sexta* Linnaeus (1763) larvae by a wasp *Cotesia glomerata* Linnaeus (1758) was found to impact larval mobility and levels of octopamine (*Adamo, 2002*). Elevated octopamine levels have been linked to increased pheromone sensitivity in moths and altered responses to non-pheromonal compounds in *Periplaneta americana* Linnaeus (1758) (*Zhukovskaya, 2012*). Despite the considerable research on how pathogens and parasites influence insect behavior, the underlying mechanisms driving changes in olfactory responses remain poorly understood (*Baverstock, Roy & Pell, 2010*). Furthermore, several areas like the chemical signals exchanged between sick insects and plants, and the impact of climate change on sick insect-plant interactions remain underexplored. Thus, further research is needed to elucidate the complex relationships between insects, plants, and diseases and inform sustainable pest management strategies.

### The Influence of Age on insects' responses to plant odor

Insects change their response to plant odors based on their age, as demonstrated by *Devaud et al. (2003)*, in their study on *Drosophila melanogaster* Meigen *(1830)*. They found that the behavioral response to benzaldehyde decreases with age, correlated with structural changes in the antennal lobe. These changes could be influenced by accumulated experiences and age-related hormone fluctuations. The attractiveness of a volatile compound is not solely determined by its chemical composition; rather, it can be influenced by factors such as the individual's physiological condition (*e.g.*, age, hormonal or mating status) and environmental conditions. For instance, in moths and locusts, behavioral responses to pheromones were altered by age and juvenile hormone levels (*Anton, Dufour & Gadenne, 2007*). While there is a significant body of research on insects' responses to plant odors, there are still many aspects that have not been thoroughly studied, particularly in the context of age. For example, the mechanisms by which older insects perceive and interpret plant odors, as well as the distinct responses of different developmental stages (such as larvae, pupae, and adults) to these odors have not been fully elucidated. Furthermore, the responses of various insect species to plant odors throughout their lifespan, as well as the impact of environmental factors such as temperature and humidity on these responses, remain largely unexplored. Finally, the molecular mechanisms underlying changes in insect responses to plant odors across their lifespan have not been fully elucidated, and practical applications of this knowledge in pest management, conservation, and agriculture are yet to be explored.

### Modifying responses to plant odor based on needs

The perception of plant odors by insects is significantly influenced by hunger and mating. When parasitoid wasps are hungry, they show a preference for the odors of flowers that serve as food sources, while fed wasps favor odors from leaves infested with host larvae for choosing oviposition sites (*Uefune et al., 2013*). Similarly, starved Colorado potato beetles and Western flower thrips exhibit heightened olfactory responses (*Davidson, Butler & Teulon, 2006*). In the case of starved *Drosophila melanogaster* flies, increased success in locating food sources is linked to neuropeptides and insulin signaling. Mating triggers a

change in olfactory preferences in female moths, potentially guiding them towards suitable oviposition sites. Mating influences the sensitivity of central olfactory neurons, with neuromodulators like biogenic amines, neuropeptides, and hormones potentially playing a role in these alterations. In the fruit fly, *Ceratitis capitata* Wiedemann *(1824)* , females' attraction to male-produced pheromone changes behaviorally after mating: mated females shift from being attracted to the sex pheromone to being attracted to plant odors (*Anton, Dufour & Gadenne, 2007*; *Jang, 1995*).These shifts in response to plant odors may involve modifications in central olfactory neurons and sensitivity of antennae (*Saveer et al., 2012*).

### Discrimination of odor stimuli

Insects have separate pathways for processing pheromone and non-pheromone olfactory stimuli, which allows for functional partitioning (*Leal, 2013*; *Mweresa et al., 2020*). This differentiation is facilitated by the differences in structure, function, and location of odor and pheromone binding proteins in the antennae and labial palps. In terms of glomerular activation patterns, odor stimuli are categorized based on their temporal and spatial characteristics (*Mweresa et al., 2020*). Many insects, like moths, tsetse flies, and mosquitoes, exhibit a flight or walking response towards the source of intermittent odor flow. The composition, concentration, and ratio of synthetic and natural odor blends are crucial in determining the selective detection and behavioral responses of insects to semiochemicals (*Grünbaum & Willis, 2015*). Selectivity at the neural level depends on various factors such as compound length, position of double bonds, and types and positions of functional groups (*Galizia & Rössler, 2010*). The speed of air flow and temporal dynamics of odor stimulation also significantly influence odor representation (*Grünbaum & Willis, 2015*). Understanding the dynamics involved in discriminating semiochemical stimuli is important for selecting and optimizing potentially behaviorally active compounds, which can be used to manipulate specific insect species (*Reisenman, Lei & Guerenstein, 2016*; *Watentena, Amos Watentena & Ikem Okoye, 2019*). These compounds may have the potential to develop novel attractant or repellent blends for insect control purposes.

## CONCLUSION AND FUTURE PROSPECTS

Insects' perception and behavioral responses to plant semiochemicals are intricately linked to their survival, reproduction, and interactions with the environment. Understanding how insects navigate through a complex milieu of chemical signals emitted by plants is crucial for elucidating the dynamics of insect-plant relationships. By unraveling the mechanisms by which insects detect and interpret semiochemicals, researchers can pave the way for innovative pest management strategies and conservation efforts. The study of plant semiochemicals and their perception and behavioral responses by insects is an exciting and rapidly evolving field in chemical ecology. Despite significant progress in understanding plant semiochemicals and insect responses, several unresolved questions and future directions exist in the field. Advancements in analytical techniques have allowed researchers to discover new plant semiochemicals and understand their roles in insect-plant interactions. Technologies like mass spectrometry, gas chromatography, and nuclear magnetic resonance spectroscopy help identify and characterize these compounds.

Studying how plants produce and release semiochemicals, as well as how insects perceive and interpret them, is crucial for understanding their ecological significance. These chemical signals can influence insect feeding behavior, oviposition, mate selection, host location, and foraging for sustenance. These chemical cues also enable insects to avoid interspecific competition, evade predation by natural enemies, and overcome the defense mechanisms of their host organisms. Discovering semiochemicals that repel or attract specific pests can lead to environmentally friendly pest control methods. However, studying semiochemical-mediated interactions is complex and involves considering other ecological factors like plant physiology, community composition, and climate change. Future research should focus on integrating multidisciplinary approaches to explore the intricate networks of chemical communication in insect-plant interactions, investigating the effects of climate change on semiochemical production and insect responses, as well as addressing challenges such as identifying novel semiochemicals and translating knowledge into practical solutions for pest control, sustainable agriculture, and biodiversity conservation.

### Funding
The author received no funding for this work.

### Competing Interests
The author declare that there is no competing interests.

### Author Contributions
- Diriba Fufa Serdo conceived and designed the experiments, performed the experiments, analyzed the data, prepared figures and/or tables, authored or reviewed drafts of the article, and approved the final draft.

### Data Availability
This is a literature review.

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
