# Peer review of "Insects' perception and behavioral responses to plant semiochemicals"

_PeerJ, doi:10.7717/peerj.17735_

## Round 0.1 · original submission · Major Revisions

The topic for review proposed by the author: plant semiochemicals and insect behavioral response, is an important subject mainly for the agricultural area with potential use in pest management and biological control. The writing is good, clear and objective. The author included good topics on the biology and physiology of insects that can be influenced in the response to semiochemicals. However, the part about types of plant semiochemicals is confusing and needs to be better structured and go into more detail. For example: volatile organic compounds (VOC's) can also be allelochemical. It's as if the author were talking about the same thing inserted into different topics. When we talk about plant semiochemistry, we include volatile and non-volatile chemical compounds. The conceptual part that talks about allelochemicals could be in the introduction and in the results there would be case studies or examples of the types of plant semiochemicals. Although some topics report examples. Case studies or examples should be in all topics for better understanding by the reader and pointing out a study for each topic covered also shows the state of research on that subject, whether there are studies or not. In general, you need to detail the topics a little more and give examples with case studies.

I ask you to carefully understand the classification of organic substances, on the one hand, and the ways in which insects react to these substances, on the other hand. Since both areas of research developed almost independently of one another, and entomologists (zoologists) and organic chemists are very distant specialties. I ask you to illustrate each of the subsections of the article with specific cases. The topic of this article is very interesting, but it has been analyzed superficially at the moment. Please do a more in-depth analysis of the problem in accordance with the comments of the reviewers.

·

Basic reporting

The study of the mechanisms of semiochemical interaction between plants and insects is an actively developing area of chemoecology. The chemical communication of many Insecta species, given the enormous species diversity of these arthropods, has been very poorly studied. The accumulation and generalization of scientific material on this topic can be used in the development of environmentally friendly, effective methods of agriculture and biodiversity conservation. Therefore, the topic of the article under review is very relevant. The manuscript is formatted according to the requirements. However, there are some shortcomings and technical comments.

1. In the “Results” section, paragraph 3.2 is missing. After subclause 3.1.2. (151) goes straight to 3.2.3. (155). Because of this, the numbering of paragraphs further in the text is disrupted.
2. The text of the manuscript contains very long sentences that are difficult for the reader to understand. I recommend breaking the sentences into at least two parts (57–60, 161–165, 376–381, etc.).
3. In subclause 3.1.1. The word "herbivores" is mentioned too often. I recommend replacing “herbivores” with the synonym “phytophagous” in some sentences.
4. In subclauses 3.4.3., 3.4.4., 3.4.5., 3.4.6. i recommend rephrasing sentences in which the action indication refers to the quotation. For example: “(Anton et al., 2007) compiled information on how insect hormones impact insect responses to both pheromones and plant volatiles…” (319–321).
5. When the Latin names of insects and plants are mentioned for the first time in the text of the manuscript, it is necessary to indicate the author’s surname and the year of description of the species (67, 68, 224, 225, etc.).
6. Species and generic Latin names of insect and plant species must be italicized.
7. When mentioned in subclause 3.4.3. the Latin name of the bacterial genus Wolbachia (304, 306), the abbreviation “sp.” must be added. (designation of an indeterminate species).
8. You need to add closing quotes before the quote (164).
9. You need to insert a space after the quote (302), and a period at the end of the sentence (197).
10. In the common nouns “Synomones” and “Flowers” located inside sentences, replace the first letters with small ones (160, 161).

Experimental design

1. Quoting the same publication twice in one paragraph and in adjacent sentences is not acceptable. One link needs to be removed from the text (346, 350).
2.It is recommended to cite literary sources at the end of the sentence.
3. In paragraphs 3.1., 3.1.1., 3.1.2., 3.2.4., 3.2.5. no citation. Needs to be added. This is a gross violation.
4. I recommend point 3.1. do not split into subparagraphs (contain little information).

Validity of the findings

No comment.

Additional comments

No comment.

Reviewer 2 ·

Basic reporting

I appreciate the author's endeavor to explore the relationship between insects' perception and behavioral responses to plant semiochemicals. While the manuscript addresses interesting aspects of insect physiology and behavior, it lacks the depth necessary for a comprehensive review. Each topic is briefly outlined, and there is a noticeable absence of in-depth analysis and discussion. Moreover, the manuscript fails to introduce any novel insights or perspectives beyond what has already been covered in existing reviews.

Experimental design

The survey method did cover a wide range of important literature or analyze existing studies thoroughly. To improve it, the author should widen his search to include more sources, both recent and classic, and critically assess their quality and relevance to ensure a complete understanding of the subject. Also, the review was not structured logically with clear paragraphs or subsections. To fix this, the author should reorganize the manuscript to create a more cohesive narrative and delve deeper into each topic.

Validity of the findings

No comment.

Additional comments

To enhance the quality of the manuscript, I recommend revisiting the content to delve deeper into the discussed topics. Providing more thorough analysis, citing recent and classic studies, and critically evaluating existing literature will enrich the manuscript and elevate it to the standards expected for publication in PeerJ.

Reviewer 3 ·

Basic reporting

Literature references, sufficient field background/context provided.

The text is a little superficial. The author needs to go into more detail about the subject, especially when discussing types of plant semiochemicals.

Experimental design

The article content is within the aims and scope of the journal nevertheless, the subsections of the result are confused and summarized.

Validity of the findings

no comment

Additional comments

The author needs to better structure the text, go into more detail about the subject, present case studies, examples in subsections.

---

## Round 0.2 · Minor Revisions

Your manuscript has improved significantly, but the mistakes pointed out by the reviewer need to be corrected. Try to make the manuscript interesting to experts in the field. It is desirable that each section of the literature review contains a new perspective on the material being studied. I would like you to draw the attention of readers not only to those features that have already been studied. It is advisable to pay more attention to phenomena that have not yet been studied. This way, your literature review will help stimulate research in this area.

Reviewer 3 ·

Basic reporting

The manuscript was better structured. The results subtopics are better divided. Within the categories of plants' semiochemicals, in the subtopics of phytochemicals and root exudation, it is necessary to mention some study that addresses plant phytochemistry and its relationship with insects. It is also necessary to include some study of exudate and insects.

On Tracked Changes:
1 - In the second paragraph of the Introduction, remove the compound methanol, as its release in plants is not common.

2- In results, in the subtopics of phytochemistry and root exudates include examples of studies that address phytochemistry and insects and root exudates and insects

3- In: Extraction, identification and characterization of Semiochemicals, Remove: which are the chemical signals that mediate interactions between organisms . This concept already explained in the introduction. There is no need to repeat the concept of semiochemical again here

4 - In: Environmental factors influencing plant odour emission and insect orientation, in the second paragraph, Remove: often described as 'volatile packages,' are released by plants and .....because
'volatile packages,' was already placed in the previous paragraph.
are released by plants, It is already understood

5 - In: Sick insects and their responses to plant odour, Phytoseiulus persimilis . This species must be written in italics and include author’s surname and the year of description of the species

Experimental design

No comment

Validity of the findings

No comment

Additional comments

Additional comments are in tracked changes.

---

## Round 0.3 · accepted · Accept

I am pleased to inform you that the article has now become much better than the original version. You have eliminated almost all the shortcomings and the article can be recommended for publication.